# Exploring Community Perspectives on Functional Paediatric Habitual Constipation

**DOI:** 10.3390/ijerph21081017

**Published:** 2024-08-02

**Authors:** Nawaf Al Khashram, Ahmad A. Al Abdulqader, Haytham Mohammed Alarfaj, Mohammed Saad Bu Bshait, Ahmed Hassan Kamal, Ossama M. Zakaria, Mohammed Nasser Albarqi, Mohannad Adnan Almulhim, Mazin Abdulwahab Almousa, Abdullah Almaqhawi

**Affiliations:** 1Department of Biomedical Sciences, College of Medicine, King Faisal University, Al Hofuf P.O. Box 400, Saudi Arabia; 2Departments of Surgery, College of Medicine, King Faisal University, Al Hofuf P.O. Box 400, Saudi Arabia; 3Department of Family and Community Medicine, College of Medicine, King Faisal University, Al Hofuf P.O. Box 400, Saudi Arabia; 4College of Medicine, King Faisal University, Al Hofuf P.O. Box 400, Saudi Arabia219041577@student.kfu.edu.sa (M.A.A.)

**Keywords:** paediatrics, habitual constipation, Saudi Arabia

## Abstract

(1) Background: Functional habitual constipation (FC) in children is a common gastrointestinal problem. This study aimed to explore the local community’s view on this problem, emphasising the challenges that parents face in managing the condition and its impact on the child’s quality of life. (2) Methods: A prospective, cross-sectional, community-based study was conducted between March and July 2023. The survey received 933 responses. The target population was adults over 18 years of age living in the Eastern Province of Saudi Arabia. An electronically distributed questionnaire was designed in the Arabic language. (3) Results: The mean knowledge scores were significantly higher in females than males, with t (931) = −2.701 and *p* = 0.007. The Bonferroni post hoc test results indicated that participants between 20 and 29 years exhibited significantly higher levels of knowledge scores compared to those between 30 and 39 years. Furthermore, the results revealed that those with three or more children had significantly higher knowledge scores compared to those with only one child. (4) Conclusions: This study demonstrated that parents in the local community have a good perceived knowledge of FC, but it needs to be linked with practice. They tended to report high levels of perception and demonstrated better practices. These results emphasise the importance of exploring the local community’s view on constipation among children.

## 1. Introduction

Functional constipation (FC) is a common gastrointestinal disorder, which is often multifactorial. It affects a significant proportion of children worldwide and can significantly impact their quality of life [1]. The prevalence of FC varies between countries, ranging from 1.3% to 26.8% [2]. The average age of its onset is between 2 and 3 years. It involves most children of preschool age, with a higher prevalence in females than males [3,4].

Diagnosis is primarily clinical, with a thorough medical history and physical examination [5]. The Rome IV criteria, which replaced the Rome III ones, have made the diagnosis of FC in children easier, with a shorter diagnostic interval [6]. The Rome IV criteria distinguish between children trained in the bathroom and those with no toilet training. If a child has not received toilet training and meets any two or more of the following criteria, they would be diagnosed with FC: fewer than two defecations per week, excessive bowel retention, painful or difficult bowel movements, large-diameter stool, and large faecal mass in the intestine (*encopresis*). Two additional criteria for toilet-trained children may include at least one episode of incontinence per week or a history of large-diameter stools that can obstruct toilets [7].

Numerous risk factors, including dietary, behavioural, psychological, medical, physical, socioeconomic, educational, and environmental factors, have been linked to paediatric FC [8]. Children from single-parent families with a positive family history, poor dietary habits, obesity, and sedentary behaviours are at higher risk of FC [9]. FC in infants may develop because of the transition from breastfeeding to formula or the introduction of solid foods [10]. Another significant risk factor is psychological stress, especially in older children. Several studies have linked FC to psychological stressors related to family and school [11]. 

Paediatric constipation constitutes a scope of different organic and functional etiologies [12]. Its origin varies according to the child’s age from the neonatal period to adolescence [13]. The passage of meconium is a crucial marker for newborn health; any delay should be promptly investigated. Sometimes, it may indicate slow transit constipation [14] and is an indicator of other abnormal colonic diseases, including *malrotation*, *colon atresia*, or *Hirschsprung’s disease*, which can be challenging to manage and require surgical intervention [15].

Functional constipation in paediatric patients can significantly impact the child and their family [16]. Additionally, it is likely to cause stool-holding behaviour, faecal impaction, psychological issues, and a lower quality of life without early treatment [17]. Almost 50% of children with constipation remain suffering, which may be attributed to the fact that *anorectal inertia* and *dyssynergia* constitute high-risk factors for persistent FC [18]. These children’s symptoms can be effectively managed by combining non-pharmacological and pharmacological interventions (i.e., *osmotic laxatives*, *polyethylene glycol*, *lactulose*, *stimulant laxatives*, *bisacodyl*, and *sodium picosulphate*) [19]. The primary treatment approaches include non-pharmacological interventions, such as education, behavioural change, diet, and reward systems [5]. However, pharmacological intervention is often required and includes *polyethylene glycol* as the first line of treatment for impact and maintenance treatment, followed by *lactulose* as an option [20].

The prevalence of FC varies widely between countries and studied regions, and few studies assess the parents’ perception of FC in the Eastern Province of Saudi Arabia. Some studies in other countries related the problem to parents’ psychological issues [21]; in local studies, they were constrained more by their knowledge of diet causes [22]. Examining the literature will reveal that certain causative factors are contingent upon the local community’s beliefs. Therefore, this study aimed to explore the local community’s view of FC, focusing on understanding the challenges faced in managing the condition and its impact on the children’s quality of life in the Eastern Province of Saudi Arabia.

## 2. Materials and Methods

This prospective, cross-sectional, community-based study was conducted from March to July 2023. It aimed to assess the knowledge, perception, and practices concerning FC in the Eastern Province of Saudi Arabia.

The sample size for a finite population (5,000,000 residents) was determined using the World Health Organization’s sample size calculator. An acceptable error rate of 5% and an expected proportion of awareness in the population of 0.5 were considered. Additionally, a type I error rate of 5% (α = 0.05) was used. The calculated sample size was 385. However, 933 participants responded to the survey. The target population was adults above 18 years and living in the Eastern Province of Saudi Arabia.

An electronic questionnaire in Arabic, addressing all the study’s specific objectives, was created and distributed to the target population after showing them the ethical approval from the research ethics committee of King Faisal University’s Deanship of Scientific Research via social media platforms, such as WhatsApp, Twitter, and Facebook, using a snowball sampling technique. The questionnaire typically explored five key areas: demographic data, general exploration of constipation, the relationship between constipation and feeding, constipation and toilet habits, and behavioural and psychologic effects of constipation.

### 2.1. Questionnaire Design

Development of the items and constructs: The survey was created in Arabic, and language experts confirmed its linguistic clarity. A comprehensive literature review was conducted via PubMed and Google Scholar to identify items that could gauge knowledge, practice, and perceptions related to FC [22,23,24,25,26,27]. The initial draft comprised 44 questions, including 12 items for assessing knowledge, 15 for evaluating proper feeding habits, 10 for gauging perception, and 7 for assessing practice (Appendix A).

The LAWSHE method was employed to gauge the survey’s content validity [28]. Opinions from five experts were sought regarding each item in the survey, and the content validity ratio (CVR) was calculated accordingly. Any question with a CVR below 0.99 was removed from the survey, eliminating four questions (one from each construct), leaving 40 items.

For the face and transitional validity, the questionnaire was distributed to 20 parents and their opinions about the clearness and readability of each item were collected and used for improvement. 

Exploratory factor analysis (EFA) was conducted to scrutinise the factorial arrangement of the questionnaire and identify the specific items that collectively formed distinct constructs. Before the analysis, the adequacy of the sampling was assessed through the Kaiser–Meyer–Olkin test [29], revealing a satisfactory overall measure of sampling adequacy of 0.731, signifying sufficient sampling. Additionally, Bartlett’s test of sphericity was employed, indicating that the correlation matrix was non-random (χ^2^ = 2503(210), *p* < 0.001).

We assumed a correlation between the factors; therefore, the oblique rotation method was employed and the data were extracted using the principal axis extraction method. 

After executing the EFA using Jamovi statistical software version 4 and incorporating the factor package, factors with loadings below 0.4 and cross-loading differences below 0.2 were eliminated. Utilising parallel analysis, eigenvalues greater than 1, and the scree plot (Figure 1), we retained four factors responsible for almost 36% of the data variance (Table 1). The first factor, labelled ‘Knowledge’, encompassed seven questions (K2, K3, K4, K5, K6, K7, and K9) designed to assess participants’ knowledge. The second factor, focusing on ‘Knowledge about proper feeding habits’, included ten items (KF1, KF3, KF4, KF5, KF6, KF7, KF9, KF10, KF11, and KF12). The third factor, termed ‘Perception of FC’, incorporated seven items (PR2, PR3, PR4, PR5, PR6, PR7, and PR8). Lastly, the fourth factor, denoted ‘practice’, comprised four items (P1, P2, P3, and P4).

Validity and reliability: The SmartPLS software version 4, developed by SmartPLS GmbH in Germany, was utilised for the partial least square structural equation modelling (PLS-SEM) method. This approach was employed to evaluate the constructs’ reliability and validity. The composite reliability (rho_c) for all the constructs surpassed the 0.7 threshold in the measurement model illustrated in Figure 2, indicating satisfactory internal consistency for the questionnaire. The convergent validity was evident, as the average variance extracted (AVE) values for all the constructs exceeded or closely approached 0.5 (Table 2).

The discriminant validity was assessed using Fornell and Larcker’s criteria [30] (Table 3). The application of consistent PLS-SEM bootstrapping for an approximate model fit revealed a standardised root mean square residual value of 0.061 (95% confidence interval = 0.054–0.072). This value, below the 0.1 threshold, indicates an acceptable fit for the model.

The responses were coded numerically from 0 to 4, with higher values assigned to responses indicating a greater level of knowledge, perception, or good practice. The total scores for each domain (knowledge of constipation, knowledge about proper feeding, perception of constipation, and toilet practice) were calculated by summing the coded responses. These total scores were then used for the inferential statistical analysis.

### 2.2. Analytical Methods

The data were statistically analysed using IBM SPSS version 25, developed by IBM Corporation (Armonk, NY, USA). The categorical variables were presented using frequency tables and percentages, while the continuous variables were represented by means and standard deviations. Pearson’s correlation test was employed for the inferential statistics to ascertain the associations between the continuous variables. On the other hand, Student’s *t*-test and a one-way analysis of variance (ANOVA) with post hoc analysis for independent samples were utilised to examine the associations between the continuous and categorical variables. A significance level of 0.05 was used to determine statistical significance for a 95% confidence interval.

## 3. Results

This study included 933 participants; most of them were female (80%) and 40.2% were over 40 years old. The majority (68%) held a bachelor’s degree, followed by less than a bachelor’s degree (19.1%), a postgraduate degree (12.4%), and uneducated (0.5%). More than half of the respondents (52.8%) had more than three children (Table 4). 

The participants’ opinions varied regarding what groups would be more constipated. Among 39% of the participants stated that constipation affected males and females equally, while about 32.6% reported not knowing. The rest suspected that it might affect boys more than girls. Regarding the importance of delayed stool passage in the first 24 h after birth as a sign of an organic problem in the gastrointestinal tract, approximately 42% of the participants believed that it was not an indicator of a possible problem (Table 5). 

When asked to assess whether statements about children’s constipation were correct or incorrect, the results indicated that the individuals’ perspectives varied slightly. Most of them (~80%) agreed that all the statements were correct, except for ‘decreasing the amount of stool’, and approximately 40% thought the statement was incorrect. Regarding the effect of birth methods on children’s constipation, 84.7% thought that caesarean birth increased the constipation frequency (Table 5).

About half of the participants agreed with the link between constipation and health problems in children (Table 5). The participants who believed in a connection between constipation and health problems were asked which specific health issues they considered. *Malnutrition* (69.0%), formula milk (53.7%), inherited causes (46.3%), *maldigestion/malabsorption* (36.7%), *neurological* disorders (31.0%), *anorectal* disorders (11.4%), *allergies* (7.2%), and *respiratory* illnesses (4.4%) were the most commonly stated (Figure 3).

When assessing the knowledge of feeding among the participants, 83.3% believed that breastfeeding reduces the risk of constipation among children. Most respondents (46.9%) believed that the ideal weaning age is between 15 and 24 months. They also believed that natural vegetables, fruits (95.6%), and yoghurt (88.8%) are suitable for weaning. Yet only 28.8% addressed whether commercial baby food was appropriate for child reabsorption. Only 52.9% agreed that children should drink at least five glasses of water every day. The vast majority of respondents (71.9%) stated that their child consumed foods like pasta and rice, which are composed of fat and carbohydrates. A large proportion (90.7%) denoted that their children have a well-balanced diet rich in vegetable and fruit fibres. Similarly, most participants (90.1%) stated their child consumes yoghurt and milk. Sweets and desserts were not consumed by 63% of the respondents’ children (Table 6).

Regarding the participants’ beliefs about practices, the majority (46.3%) allowed sweet and dessert consumption more than once a day, and only 24% allowed it 3–6 times per week, but the recommended answer was ‘less than twice a week’, which none of the participants selected (0%). A considerable percentage of respondents (39.9%) intended to stop using diapers for their children between 24 and 36 months. Regarding the frequency of toilet bowel habits, the majority (53.8%) reported more than four times a week. When asked if they ever utilised any kind of aid with their kids’ bowel motions, 81.6% said they never did, while 10.1% used medicine or suppositories, and 8.4% used natural remedies or products. The participants responded differently about the pain frequency during their children’s bowel motions: few participants answered always (3.8%) and often (13%), while many reported sometimes (57.6%) or never (25.7%). Various levels of refusal to use the toilet outside the home for bowel movements were reported by the participants for their children. Most of them said their children refused sometimes (42.2%) or never (31.1%). When it came to clothes-soiling with stool during constipation, some participants said it always happened (6.1%), and the majority said it happened sometimes (42.2%) or never (37.1%) (Table 7).

In response to the questions regarding the effects of constipation, 47.8% of the participants agreed that it could negatively affect a child’s overall health, while 47.8% agreed that it could impact their behaviour. Moreover, 37.8% were neutral about the effects on academic performance, and 44.9% agreed that it could decrease appetite or lead to weight loss (Figure 4).

The minimum score for the ‘Knowledge about constipation’ was 9, the maximum was 20, and the mean score was 15.76 ± 2.02. The independent samples *t*-test indicated a statistically significant difference in the mean knowledge scores between males (15.41 ± 2.25) and females (15.85 ± 1.94) [t (931) = −2.701, *p* = 0.007], with a small effect size (Cohen’s D = −0.21). Specifically, females achieved a significantly higher mean knowledge score compared to males. Furthermore, an ANOVA was conducted to examine the relationship between the number of children groups and the knowledge scores. The results revealed a statistically significant difference in the mean scores [F (3929) = 5.773, *p* = 0.001], with a small effect size (η^2^ = 0.011). The results of the Bonferroni post hoc test indicated that participants who had three or more children (15.98 ± 2.18) exhibited significantly higher knowledge scores than those with only one child (15.23 ± 1.99) (*p* = 0.001).

The ‘Knowledge about feeding’ had a minimum score of 17, a maximum of 30, and a mean of 23.21 ± 2.05. A statistically significant difference was observed among the different age groups, as determined by ANOVA [F (2930) = 4.023, *p* = 0.018], with a small effect size (η² = 0.009). The results of the Bonferroni post hoc test indicated that individuals within the age range of 20–29 years (23.50 ± 2.09) exhibited significantly higher knowledge scores than those between 30 and 39 years (23.03 ± 2.3) (*p* = 0.017).

The ‘Perception about FC’ scores displayed a minimum of 4, a maximum of 20, and a mean score of 14.76 ± 2.84. ANOVA also revealed a statistically significant difference in the mean scores for ‘practice of toilet training’ among the groups with different educational levels [F (3929) = 5.295, *p* = 0.001], with a small effect size (η^2^ = 0.017). The Bonferroni post hoc test indicated a significant difference in the practice scores between the participants who possessed a postgraduate degree (15.45 ± 2.75) and those with less than a bachelor’s degree (14.22 ± 3.25), with the former group exhibiting higher scores (*p* = 0.001). The ‘practice of toilet training’ score ranged between 12 and 30, with a mean score of 23.03 ± 2.86. No significant differences were observed in the mean scores among the different respondent groups.

Pearson’s correlation test revealed a statistically significant positive correlation between the knowledge scores and the perception and practice scores. As the knowledge scores increased, the participants tended to report higher levels of perception (*r* = 0.172, *p* < 0.001) and demonstrated better practice scores (r = 0.196, *p* < 0.001).

## 4. Discussion

FC is a highly encountered problem among children [31] and has been thoroughly discussed in the literature, but rarely holistically [22]. A local study with a similar setting indicated that it affected more than a quarter of the national paediatric population [32]. Despite the high prevalence of FC among the national community, parental perception has been assessed in other studies more concerning the causes and symptoms of nutritional constipation; they did not consider additional factors that may contribute to non-food-related constipation, including certain medical conditions, the optimal age for transitioning out of diapers, and prolonged avoidance of using public restrooms [33].

This qualitative cross-sectional study was designed to evaluate local parents’ perception of this problem, with a special emphasis on the familial, nutritional, and functional factors, such as toilet training, in relation to geographical and sociodemographic particulars. Most participants in the current study were females, which coincided with previously published studies from the same national setting [22,23,25,32]. This prevalence of females may be attributed to the fact that mothers are more involved in rearing children than their male peers. We noticed that participants’ responses to the causes of constipation were limited to food only. Individual dietary intake and impairment of digestive and absorption processes, as illustrated in Figure 3, are highlighted while ignoring the consideration of other organic, genetic and psychological disorders, which have been extensively investigated in several other studies [34]. 

During this study, we observed that the existence of correct knowledge sometimes interfered with the practices or beliefs; an example of this was when we asked about the use of sweets and sugars in the diet for 2 years of age, where 63.03% of them said no. This shows that knowledge is good among participants in the study. However, when we asked about their practices with children and how many sugars and desserts were given during the week, 0% said ‘less than two times a week’. By contrast, 46.3% of respondents provided the worst response of ‘more than once a day’. This proves that knowledge can only be sufficient by linking it to proper practice or belief. Therefore, we consider it important that the community is medically educated on some diseases, such as constipation, by providing knowledge in combination with the practices and beliefs associated with them.

Moreover, more than half of the participants were between 30 and 50 years old [22,23,25,32]. Women in this age group are usually multipara and have previously encountered the problem with their older children. Additionally, most participants were well-educated university graduates who were knowledgeable about the problem through their professional careers. Another reason could be the accessibility of various respectable social media information, which contradicts the low information among younger age groups with lower educational profiles [22,23,25,32]. Although maternal employment is a highly influential factor in the predisposition to paediatric FC due to mothers’ professional commitments, the current study did not discuss this concept [23]. This factor may contribute to a higher risk of FC among children [23].

Nevertheless, a considerable percentage of the study respondents expressed ignorance or lack of knowledge about the gender factor of FC. This result agrees with previously published data from a lower developing setting with a comparatively lower socioeconomic status [13]. 

A previously published local study reported a moderate level of knowledge regarding the causes of constipation among children [22], while others reflected poor knowledge. They demonstrated a statistically significant relationship between good knowledge and good practices regarding dealing with childhood constipation [23]. This result emphasises the importance of exploring parents’ level of perception regarding their children’s constipation. 

Most participants displayed a relatively good view of the FC causes, including a lack of frequency, hard constancy, and some symptoms that may accompany them. They also added bloating and pain. However, some believed there was no relationship between constipation and the amount of stool. Many authors contradicted these false beliefs [35]. Children might suffer from FC because of their parents’ ignorance. The latter might think that their children are healthy as long as they visit the toilet daily and the constancy of their stool is normal, even though the amounts are minimal. Lower amounts could be due to infrequent or incomplete stool evacuation, one of FC’s important characteristics [36]. About half of the current study participants stated that the optimal age for weaning is between 15 and 24 months. This answer contradicted others, who stated a lower age of weaning, with a low-fibre diet as the etiological cause of FC [27]. These data were supported by similar national studies [23]. 

The people with multiple children, between 20 and 29 years old who had completed their education up to and including a bachelor’s degree provided the most accurate answers regarding their knowledge, practice, and beliefs. This further emphasises the significance of our research, as it demonstrates that the correct knowledge, practice, and beliefs are essential for managing constipation, a discovery that has not been documented in previous studies. 

### Strengths and Limitations of the Study

Regarding the knowledge and practice towards FC, no difference was found between the current study and similar national studies. However, the current study found that linking correct knowledge with practice and belief is essential for dealing with constipation. This was not mentioned in other studies. This study has a relatively limited number of participants compared to the whole population. Additionally, some parents lacked experience with children, which may introduce biases into the research results. Consequently, further qualitative studies may be needed to follow the problem and introduce a realistic national view of the FC problem by comparing the perspective between the parents of known constipation patients and the community.

## 5. Conclusions

This study demonstrated that local parents have a positive perception of their knowledge regarding FC, but it needs to be linked with practice. The overall results and the other national studies highlighted the need for a strong national social educatory consensus to deal with FC problems in the community. This entails more focused national health programmes to be addressed via social media and other public educatory tools to overcome the shortage of proper knowledge, practice, and belief towards FC across the nation. Further educational programmes must be established to evaluate the problem among older children and adolescents in the community. More intensive efforts should be committed to involving extra knowledge, practice, and proper attitude towards FC among the school curricula in different paediatric age groups. Such practices should run from kindergarten up to high school. The current study supports the concept that a high education level would be the main method for fighting FC.

## Figures and Tables

**Figure 1 ijerph-21-01017-f001:**
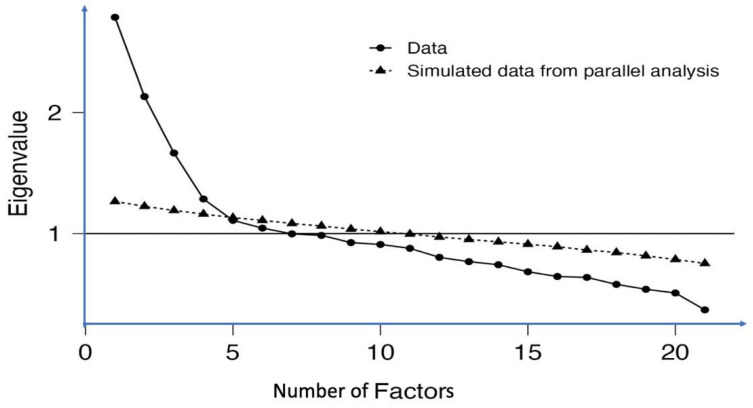
Scree plot generated through parallel analysis. All four retained factors are positioned above the intersection point of the actual and simulated data.

**Figure 2 ijerph-21-01017-f002:**
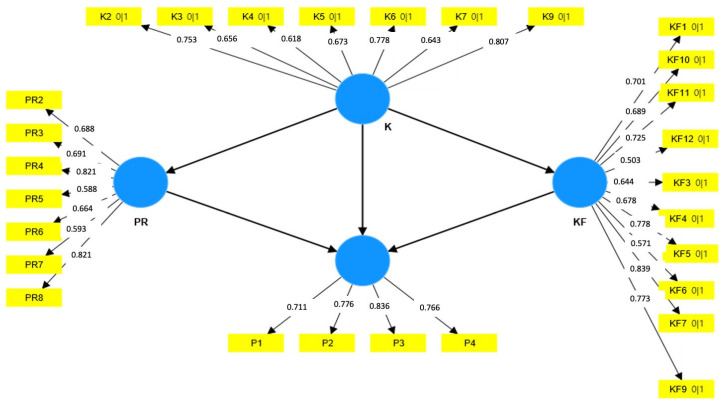
Measurement model and factor loadings associated with each questionnaire construct. Yellow rectangles symbolise the questions (factors and items), while blue circles depict the constructs (latent variables and domains).

**Figure 3 ijerph-21-01017-f003:**
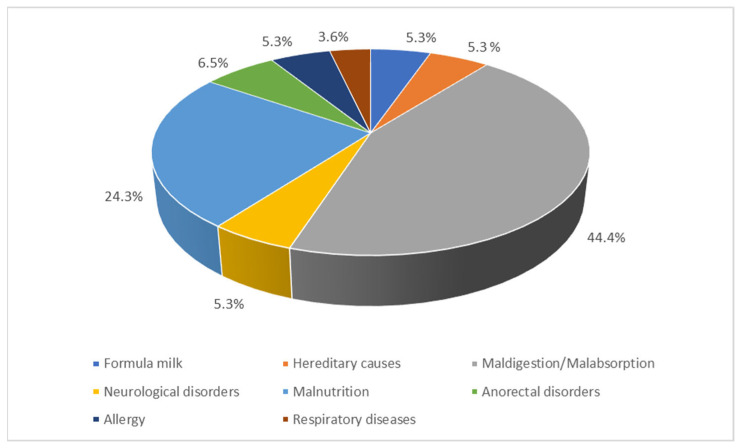
Health issues connected to constipation, according to the participants (N = 933).

**Figure 4 ijerph-21-01017-f004:**
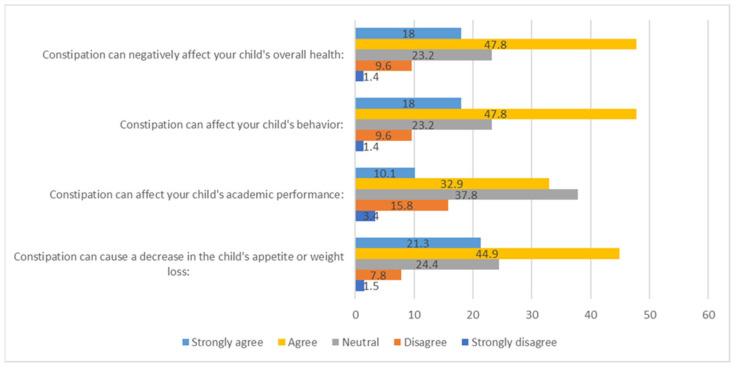
Participants’ responses about their perceptions of the impacts of constipation on organic and psychological health.

**Table 1 ijerph-21-01017-t001:** Item loadings of the different factors and their proportional variance.

Questions	Knowledge	Knowledge of Feeding	Practice	Perception	Uniqueness
K2	0.753	−0.003	0.088	−0.116	0.842
K3	0.656	−0.07	0.062	−0.076	0.729
K4	0.618	−0.013	0.053	0.001	0.648
K5	0.673	−0.051	0.055	−0.075	0.678
K6	0.778	0.09	0.135	0.047	0.511
K7	0.643	−0.046	0.034	−0.056	0.523
K9	0.807	0.033	0.199	−0.045	0.816
KF1	0.095	0.701	0.162	0.083	0.807
KF3	0.014	0.689	0.008	0.041	0.527
KF4	0.069	0.725	0.069	0.02	0.715
KF5	0.091	0.503	0.002	−0.006	0.484
KF6	0.1	0.678	0.014	−0.077	0.802
KF7	0.029	0.678	−0.13	−0.146	0.475
KF9	0.07	0.778	0.006	−0.103	0.583
KF10	0.066	0.571	0.055	0.097	0.712
KF11	0.029	0.839	0.041	0.119	0.592
KF12	−0.038	0.773	−0.002	0.185	0.447
PR2	0.05	0.22	0.06	0.688	0.666
PR3	−0.026	0.194	0.031	0.691	0.535
PR4	−0.044	0.139	−0.032	0.821	0.482
PR5	−0.088	0.07	−0.062	0.588	0.702
PR6	−0.116	0.131	−0.126	0.664	0.553
PR7	0.085	−0.053	0.214	0.593	0.688
PR8	−0.003	−0.167	0.023	0.821	0.828
P1	0.128	0.091	0.711	−0.067	0.503
P2	0.135	0.022	0.776	−0.102	0.427
P3	0.115	0.045	0.836	−0.088	0.327
P4	0.112	0.127	0.766	−0.015	0.42
Proportional variance	0.11	0.078	0.081	0.09	Cumulative
					0.359

**Table 2 ijerph-21-01017-t002:** Composite reliability (roh_c) and AVE values used to examine the internal consistency and convergent validity.

	AVE	Composite Reliability (roh_c)
Knowledge	0.501	0.87
Knowledge of feeding	0.490	0.90
Perception	0.503	0.88
Practice	0.598	0.86

**Table 3 ijerph-21-01017-t003:** Fornell and Larcker’s criteria used to examine the discriminative validity.

	Knowledge	Knowledge of Feeding	Practice	Perception
Knowledge	0.707			
Knowledge of feeding habits	−0.026	0.701		
Practice	0.158	0.091	0.773	
Perception	−0.086	0.246	−0.088	0.709

**Table 4 ijerph-21-01017-t004:** Demographical data of the study participants.

Variable		Frequency	Percentage
Gender			
	Male	187	20.0
	Female	746	80.0
Age			
	20–29 years old	262	28.1
	30–39 years old	296	31.7
	Over 40 years old	375	40.2
Education			
	Uneducated	5	0.5
	Less than a bachelor’s degree	178	19.1
	Bachelor’s degree	634	68.0
	Postgraduate degree	116	12.4
No. of Children			
	No children	150	16.1
	One child	137	14.7
	Two children	153	16.4
	More than three	493	52.8

**Table 5 ijerph-21-01017-t005:** Participants’ general knowledge of constipation.

Question	Answer	Frequency	Percentage
Constipation is more common in			
	I do not know	304	32.6
	Male	120	12.9
	*Female*	145	15.5
	Equally	364	39
Is delayed passage of stool during the first 24 h after birth a sign of an organic problem in the intestines?			
	No	393	42.1
	*Yes*	540	57.9
Which of the following statements regarding constipation in children are correct?			
- Infrequent bowel movements (once every three days or more)			
	Incorrect	191	20.5
	*Correct*	742	79.5
- Increased stool hardness and change in shape			
	Incorrect	151	16.2
	*Correct*	782	83.8
- Reduced stool quantity			
	Incorrect	372	39.9
	*Correct*	561	60.1
- Severe pain and difficulty in defecation			
	Incorrect	103	11
	*Correct*	830	89
- Frequent bloating and increased gas output			
	Incorrect	192	20.6
	*Correct*	741	79.4
Do delivery methods affect constipation in children?			
	Constipation increases with vaginal delivery	8	0.9
	*Constipation increases with caesarean delivery*	790	84.7
	There is no difference	135	14.5
Does constipation in children relate to health problems?			
	No	449	48.1
	*Yes*	484	51.9

Correct answers are in italics.

**Table 6 ijerph-21-01017-t006:** Knowledge about dietary materials and their relation to constipation.

Question	Answer	Frequency	Percentage
Does breastfeeding reduce the chances of your child experiencing constipation?			
	No	156	16.7
	*Yes*	777	83.3
What is the optimal age to wean your child to prevent constipation?			
	More than 24 months	156	16.7
	15–24 months	438	46.9
	12–15 months	166	17.8
	9–12 months	106	11.4
	*6–9 months*	67	7.2
What is the best food for weaning your child?			
- Natural vegetables and fruits			
	No	41	4.4
	*Yes*	892	95.6
- Yoghurt			
	No	142	15.2
	*Yes*	791	84.8
- Ground grains (e.g., cereals)			
	No	289	31
	*Yes*	644	69
- Boiled meat			
	No	501	53.7
	*Yes*	432	46.3
- Commercial baby food			
	No	664	71.2
	*Yes*	269	28.8
Your child needs to drink at least five cups of water daily (one cup equals 200 mL)			
	No	172	18.5
	I don’t know	268	28.7
	*Yes*	493	52.9
What kind of food does your child over two years old eat?			
- Food containing fats and carbohydrates, such as rice and pasta			
	No	262	28.1
	*Yes*	671	71.9
- A balanced diet containing vegetable and fruit fibres			
	No	87	9.3
	*Yes*	846	90.7
- Yogurt and milk			
	No	92	9.9
	*Yes*	841	90.1
- Sweets and desserts			
	Yes	342	36.7
	*No*	591	63.3

Correct answers are in italics.

**Table 7 ijerph-21-01017-t007:** Practice of toilet training and habits.

Question	Answer	Frequency	Percent
How often do you allow your child to eat sweets and desserts:			
	More than once a day	432	46.3
	Once a day	277	29.7
	3–6 times a week	224	24
	*Less than twice a week*	0	0
I will start by stopping the use of diapers for my children at the age of:			
	More than 36 months	218	23.4
	24–36 months	30	3.2
	24–36 months	372	39.9
	12–15 months	115	12.3
	*15–24 months*	198	21.2
How often does your child have a bowel movement in the toilet:			
	Less than once a week	53	5.7
	1 to 2 times a week	92	9.9
	3 to 4 times a week	286	30.7
	*>4*	502	53.8
Does your child use any aids to help with bowel movements:			
	Yes, medication/suppositories	94	10.1
	Yes, natural remedies/products	78	8.4
	*No*	761	81.6
Does your child suffer from or feel pain during bowel movements:			
	Always	35	3.8
	Often	121	13
	Sometimes	537	57.6
	*Never*	240	25.7
Does the child refuse to use the toilet outside the home (public places and school) for bowel movements:			
	Always	64	6.9
	Often	185	19.8
	Sometimes	394	42.2
	*Never*	290	31.1
Do the child’s clothes get soiled with stool when they suffer from constipation:			
	Always	57	6.1
	Often	136	14.6
	Sometimes	394	42.2
	*Never*	346	37.1

Good practice responses are in italics.

## Data Availability

The datasets used and analysed during the current study are available from the corresponding author upon reasonable request.

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
