# Peer review of "Exploring Community Perspectives on Functional Paediatric Habitual Constipation"

_ijerph, 2024, doi:10.3390/ijerph21081017_

Round 1
Reviewer 1 Report
Comments and Suggestions for Authors
Manuscript entitled "Pediatric Habitual Constipation: A Local Community View" by Nawaf Al Khashram, Ahmad A. Al Abdulqader, Haytham Mohammed Alarfaj, Mohammed Saad Bubshait, Ahmed Hassan Kamal, Ossama M. Zakaria, Mohammed Nasser Albarqi, Mohannad Adnan Almulhim, Mazin Abdulwahab Almousa, and Abdullah Almaqhawi
The manuscript provides a valuable perspective on the management of functional pediatric habitual constipation, focusing on parental perceptions within a local community. This study is particularly relevant due to the high prevalence and significant impact of pediatric constipation on children's quality of life. The comprehensive analysis of parental knowledge and practices regarding constipation, supported by robust statistical validation of the questionnaire, makes a notable contribution to the existing literature.
However, there are several issues that need to be addressed to enhance the manuscript:
- The concluding part of the introduction lacks a clear statement distinguishing what is already known from what your study adds to the literature.
- The explanation of the sample size calculation is brief and lacks references to support the methodology used. Expanding this section to include a detailed rationale and references to similar studies or standard texts on survey methodology would strengthen the validity of the research design.
- Displaying the questionnaire, or at least a representative section, in English could enhance the transparency of the study and allow for broader scrutiny of the tool used, particularly regarding its relevance and comprehensiveness.
- The manuscript describes demographic data collection but does not detail how these data were statistically analyzed.
- The recruitment process and criteria for participant selection are not thoroughly described.
- While some statistical tests are mentioned, the manuscript would benefit from a more detailed explanation of the statistical methods used, especially how they relate to the study questions and data structure.
- The discussion section is overly concise and does not sufficiently explore the broader implications of the findings. Expanding this section to discuss how the results integrate with or challenge existing clinical practices or policies could provide more depth.
- The manuscript does not critically address the limitations of the study or the potential biases in sampling methods and data collection.
- Several typographical and grammatical errors throughout the text could detract from its professional quality and readability. Comprehensive proofreading and editing are recommended to enhance the overall presentation.
Comments on the Quality of English Language
- Several typographical and grammatical errors throughout the text could detract from its professional quality and readability. Comprehensive proofreading and editing are recommended to enhance the overall presentation.
Author Response
Thanks for sending the detailed review comments for this manuscript to us to revise for consideration again. We appreciate your comments and suggestions. We have responded to the comments listed in the attached file.

Reviewer 2 Report
Comments and Suggestions for Authors
The manuscript by Nawaf Al Khashram et al. provided a local community view about the pediatric habitual constipation. The study is informative and interesting. I have the following questions and comments:
1, all the tables should be in a three-line format.
2, the y axis in figure 1 must be provided and revised.
3, in figure 3, the numbers should be 24.3 %, 44.4 % etc. The authors must revise.
4, figure 4 is poorly prepared. The x axis is missing. The y axis is missing. Figure labels and legend are not appropriate. The authors must revise.
5, The English writing must be further improved.
Author Response
Thanks for sending the detailed review comments for this manuscript to us to revise for consideration again. We wish to thank you for your comments and suggestions. We have responded to the listed comments as below.
Comments 1: all the tables should be in a three-line format.
Response 1 : According to the reviewer, all the tables have changed the format.
Comments 2: the y axis in figure 1 must be provided and revised.
Response 2 : The figure 1 has been updated.
Comments 3: in figure 3, the numbers should be 24.3 %, 44.4 % etc. The authors must revise.
Response 3 : Thanks for the suggestion. We modified that according to the reviewer suggestion.
Comments 4: figure 4 is poorly prepared. The x axis is missing. The y axis is missing. Figure labels and legend are not appropriate. The authors must revise.
Response 4 : The figure 4 has been updated.
Comments 5: The English writing must be further improved.
Response 5 : We checked it according to the reviewer's comment.
Reviewer 3 Report
Comments and Suggestions for Authors
The manuscript entitled” Pediatric Habitual Constipation: A Local Community View “highlights the importance of exploring the local community view of constipation among children. Lack of clarity at some points in abstract and conclusion is visible. In addition, a careful grammar revision is advised. Finally, after careful examination, I appreciate the quality of work. I have some concerns regarding the clarity and detail, and precision of different sections
· The title is not reflecting the true sense-it is looking a review paper. Please reconsider as per the aim and objective of research
· Abstract- Sentence selection is very informal. Keep a proper sequence in the abstract and end with conclusion. Focus on the Research gap you have covered in our study What? Why? How? Findings of the study? Conclusion is very poor.
· L-16-17 Reconsider- Abstract – A prospective, cross-sectional, community-based study took place during a six 16 months period from March 2023 to July 2023. An electronically distributed questionnaire was de- 17 signed in Arabic language
· L-45 Complete this sentene- A range of risk factors for pediatric FC have been identified [8]
· L-65Please make it rationalr- There is a wide difference in the prevalence of FC between countries and stud- 65 ied regions, as well as a lack of studies assessing the parents’ perception as regards FC in 66 the local community. Some of the studies in other countries related the problem more with 67 parents' psychological issues [17], in local studies they constrained more on their 68 knowledge [18]. When we look through these literatures, we will find that some of the 69 causal factors are variable according to local community beliefs. Therefore, this study took 70 place to explore the local parental view of functional pediatric habitual constipation, em- 71 pathizing on understanding the challenges they face in managing the condition and its 72 impact on the children’s quality of life
· L-83-84 Any mapping /selection criteria ??????????????????? This prospective, cross-sectional, community-based study was carried out in the pe- 75 riod from March 2023 to July 2023. It aimed to assess the knowledge, attitude, and Practice 76 of parents for functional pediatric habitual constipation within the local community. 77 The minimum required sample size of 385 was calculated using the sample size cal- 78 culation formula with a margin of error of 5%, a confidence level of 95%, and a population 79 size of 5,000,000." 80 An electronic questionnaire was designed in Arabic language covering all the specific 81 objectives of the study, and then it was distributed electronically to the target population 82 (Parents of school and preschool children) who were encouraged to fill the questionnaire. 83 The questionnaire typically included five key elements. They were demographic data re- 84 lated questions, exploration of constipation in general, questions about the relationship 85 between constipation
· Any ethical comitte approvel ??????????????? When asked to assess whether; statements about children's constipation were correct or 183 incorrect, the results showed that individuals' perspectives varied slightly. The majority 184 (approximately 80%) agreed that all statements were correct, except for "decreasing the 185 amount of stool". Approximately, 40% thought the statement was incorrect. Concerning 186 how birth methods might have affected child constipation, 84.7% thought that cesarean 187 birth lead to increase constipation frequency, (Table 5
· Nned clarity- Regarding the practice and attitude towards FC, there were no distinct difference 327 between the current study and similar national studies. The number of the study 328 participants is relatively limited compared to the total population. Therefore, further 329 qualitative studies may be needed to follow up the problem and introduce a realistic 330 national view about FC problem
· Cite the following referecns-
§ Childhood constipation as an emerging public health problem. World journal of gastroenterology, 22(30), p.686
§ . Perceptions, definitions, and therapeutic interventions for occasional constipation: A Rome working group consensus document. Clinical Gastroenterology and Hepatology, 22(2), 397-41
· Italic all the scientific names,
· Remove grammatical mistakes
· Need to rewrite the conclusion
§ Recheck Legends description is as per figure number and discussion-
§ I urge the authors to improve the English language for better flow of literature.
§ Please check reference style throughout MS
§
Comments on the Quality of English LanguageModerate editing of English language required
Author Response
Thanks for sending the detailed review comments for this manuscript to us to revise for consideration again. We wish to thank you for your comments and suggestions. We have responded to the listed comments as below.
Comments 1: The title is not reflecting the true sense-it is looking a review paper. Please reconsider as per the aim and objective of research.
Response 1: We modified that to : Exploring Community Perspectives on Functional Paediatric Habitual Constipation
Comments 2: Abstract- Sentence selection is very informal. Keep a proper sequence in the abstract and end with conclusion.
Response 2: Thank you we will consider that after the proofreading.
Comments 3: Focus on the Research gap you have covered in our study What? Why? How? Findings of the study? Conclusion is very poor.
Response 3: Thanks for the suggestion. We modified that according to the reviewer suggestion.
Comments 4: L-16-17 Reconsider- Abstract – A prospective, cross-sectional, community-based study took place during a six 16 months period from March 2023 to July 2023. An electronically distributed questionnaire was de- 17 signed in Arabic language
Response 4: Thanks for the suggestion. We modified that according to the reviewer suggestion.
Comments 5: L-45 Complete this sentene- A range of risk factors for pediatric FC have been identified [8
Response 5: Thanks for the suggestion. We modified that according to the reviewer suggestion.
Comments 6:
- L-65Please make it rational- There is a wide difference in the prevalence of FC between countries and stud- 65 ied regions, as well as a lack of studies assessing the parents’ perception as regards FC in 66 the local community. Some of the studies in other countries related the problem more with 67 parents' psychological issues [17], in local studies they constrained more on their 68 knowledge [18]. When we look through these literatures, we will find that some of the 69 causal factors are variable according to local community beliefs. Therefore, this study took 70 place to explore the local parental view of functional pediatric habitual constipation, em- 71 pathizing on understanding the challenges they face in managing the condition and its 72 impact on the children’s quality of life
Response 6: Thanks for the suggestion. We modified that according to the reviewer suggestion.
Comments 7: L-83-84 Any mapping /selection criteria ??????????????????? This prospective, cross-sectional, community-based study was carried out in the pe- 75 riod from March 2023 to July 2023. It aimed to assess the knowledge, attitude, and Practice 76 of parents for functional pediatric habitual constipation within the local community. 77 The minimum required sample size of 385 was calculated using the sample size cal- 78 culation formula with a margin of error of 5%, a confidence level of 95%, and a population 79 size of 5,000,000." 80 An electronic questionnaire was designed in Arabic language covering all the specific 81 objectives of the study, and then it was distributed electronically to the target population 82 (Parents of school and preschool children) who were encouraged to fill the questionnaire. 83 The questionnaire typically included five key elements. They were demographic data re- 84 lated questions, exploration of constipation in general, questions about the relationship 85 between constipation
Response 7: The Selection criteria has modified according to the reviewer's comment.
Comments 8: Any ethical comitte approvel ??????????????? When asked to assess whether; statements about children's constipation were correct or 183 incorrect, the results showed that individuals' perspectives varied slightly. The majority 184 (approximately 80%) agreed that all statements were correct, except for "decreasing the 185 amount of stool". Approximately, 40% thought the statement was incorrect. Concerning 186 how birth methods might have affected child constipation, 84.7% thought that cesarean 187 birth lead to increase constipation frequency, (Table 5
Response 8: The study proposal and questionnaire has an ethical approved from the research ethics committee of King Faisal University's Deanship of Scientific Research, and it showed to the participants before answering the questionnaire.
Comments 9: Need clarity- Regarding the practice and attitude towards FC, there were no distinct difference 327 between the current study and similar national studies. The number of the study 328 participants is relatively limited compared to the total population. Therefore, further 329 qualitative studies may be needed to follow up the problem and introduce a realistic 330 national view about FC problem
Response 9: Thanks for the suggestion. We modified that according to the reviewer suggestion.
Comments 10:
Cite the following referecns-
- Childhood constipation as an emerging public health problem. World journal of gastroenterology, 22(30), p.686
§ . Perceptions, definitions, and therapeutic interventions for occasional constipation: A Rome working group consensus document. Clinical Gastroenterology and Hepatology, 22(2), 397-41
Response 10: Thanks for the suggestion. We cited that according to the reviewer suggestion.
Comments 11: Italic all the scientific names
Response 11: Thanks for the suggestion. We modified that according to the reviewer suggestion.
Comments 12: Remove grammatical mistakes
Response 12: We checked it according to the reviewer's comment.
Comments 13: Need to rewrite the conclusion
Response 13: Thanks for the suggestion. We modified that according to the reviewer suggestion.
Comments 14: Recheck Legends description is as per figure number and discussion
Response 14: We checked it according to the reviewer's comment.
Comments 15: I urge the authors to improve the English language for better flow of literature.
Response 15: We checked it according to the reviewer's comment.
Comments 16: Please check reference style throughout MS
Response 16: We checked it according to the reviewer's comment.
Comments 17: Moderate editing of English language required
Response 17: We checked it according to the reviewer's comment.
Reviewer 4 Report
Comments and Suggestions for Authors
General comment:
This is a prospective, cross-sectional, study that looked at the local population’s understanding of functional habitual constipation through electronic questionnaires. The methodology lacked some essential details, such as the settings (see specific comments). Generally speaking, the methodology needs to be specific that other researchers can repeat the study based on the written information provided.
It is unclear what the clinical significance of this study is. It needs to be clear what new knowledge this study is bringing to the scientific community. E.g. what are the big picture the authors want to tell us now that they found that there are difference in knowledge scores in different age and gender. The authors also commented they found no distinct difference between the current study and similar national studies (line 327-328). Then, what is innovative about this study?
The authors stated in the introduction (line 70-73) that the study aimed to empathize on understanding the challenges parents face in managing the condition and its impact on the children’s quality of life. But the study data seemed to be only focusing on the participants’ level of knowledge. The abstract also stated the study data represented the views of the parents of the local community. But there is a lack of description of what local community this study was investigating.
Some of the participants here were apparently not parents as they declared having no children (n = 150) in Table 4 (line 174). But author’s study aim (line 70-73) stated the study aim was to explore the local parental view of functional pediatric habitual constipation. This could be a serious flaw in this study as the results need to be supportive of the study aim and conclusion. If the authors want to maintain their objective of exploring the local parental view, major revisions are needed. E.g. removing the 150 participants from the 933 total participants and re-analyzed all the data.
Some of the statistical comparison were very intriguing in which the two comparison groups had virtually the same scores, but generated a significant p-value. E.g. Line 251-252 (15.41 vs. 15.85 with huge error bars but generated a p value of 0.0120; Line 243-244 (23.50 vs. 23.03, with huge error bars, but the p-values were very significant at 0.017. Although the raw data does not need to be published, perhaps, this should be submitted to the reviewers and editor, with description of the statistical test used, to demonstrate the validity of these statistical comparison.
Specific comments:
Abstract
Line 15-18: Generally speaking, the abstract should describe at least the sample size and setting (Saudi Arabia - which local community?).
Line 18: Perhaps, it is best to clarify whether the females or males scored higher to prevent confusion and misinterpretation.
Line 23-26: It said the local community already have a good perceived knowledge about functional pediatric habitual constipation. Then why is it important to explore the local community view of constipation among children?
Introduction
Line 40-41: Consider using colon rather than semicolon after “The criteria is” e.g. “The criteria are the followings: “
Line 55-56: The part about “remain suffering” need to be more concise. E.g. how long are they remaining to suffer? What are they suffering on? Psychological or physical suffers?
Line 61: “This” often requires pharmacological intervention. Can you clarify what “this” means?
Line 62-64: Because guidelines from different areas differ. When you talk about first- and second-line treatment, it is best to be explicit on which guidelines recommend that, rather than having a reference only. It seems like reference 16 is based on a literature review conducted by two authors only, but not from an authoritative guideline.
Line 69-70: “When we look through these literatures, we will find that some of the causal factors are variable according to local community beliefs.” I am not sure what purpose this sentence serves. Or are you saying you have already looked at these factors and now want to summarize the findings? If this sentence does not add anything, please consider removing it.
Methods
Line 76: consider using all lower case for “Practice”
Line 80: what is the “ at the end of the sentence for? It is important to reference or mention which sample size calculation you used. It is unclear why you said a population size of 5 million but a minimum sample size of 385.
Line 81: To enhance the reproducibility of a study, the methodology can include the original Arabic language questionnaires and their English translation in the Appendix.
Line 83: How are the target population (parents) reached? Do you mean all the school and preschool parents in Saudi Arabia, or only in a certain region? At a minimum, the setting of the research needs to be described. Were there rewards / compensation given to the participants? What are the inclusion and exclusion criteria for participants?
Line 90: The part about literature review to identify items seem a bit vague. Do you mean you adapted the survey questions from references 18-23?
Line 95-98: The LAWSHE method and Content Validity Ratio should be referenced in case readers want to check the validity of the methods.
Line 103-111: Similarly, the Exploratory factor analysis, Kaiser–Meyer–Olkin test, oblique rotation method (Oblimin), and Principal Axis Extraction method should be referenced.
Line 138: The Fornell and Larcker criteria should be referenced.
Results
Line 171: Perhaps, for the readers who are unfamiliar with the Saudi Arabia education system, can you explain what “uneducated” mean (either in results or methodology)? Do you mean no sort of education whatsoever since childhood?
Line 174: “one, two, and more” should have the first letter capitalized like the other rows
Line 174: Some of the participants here were apparently not parents as they declared no children. But author’s study aim (line 70-73) stated the study aim was to explore the local parental view of functional pediatric habitual constipation.
Line 185: Why is there a semicolon after “whether”?
Line 190 (Table 5): It is difficult for readers to visualize what is supposed to be the correct response in each question.
E.g. constipation is more common in females/male/equally?
E.g. Delayed passage of stool during the first 24 hours after birth is a sign of an organic problem in the intestines: Is yes or no the correct answer?
E.g. Infrequent bowel movements (once every 3 days or more): What is the correct answer here?
Please consider remaking this entire table for clarity.
Line 199-200 (Figure 3): Generally speaking, a figure in a journal should be self-explanatory on its own. Are these numbers percentages? The number of respondents (n) should be stated in the figure title or legend.
Line 214 (Table 6): Like the problem in Table 5, it is difficult for readers to visualize what is supposed to be the correct response in each question. Please consider remaking this entire label for clarity. It also means some rewriting of Lines 201-212 for clarity.
Line 218: Why is there a comma after “allowed”?
Line 232 (Table 7): Like the problems in Table 5 and 6, what is supposed to be the correct response here? How are these responses relevant to functional habitual constipation?
Line 239-240 (Figure 4): Like line 199-200 (figure 3): Are these numbers percentages? The number of respondents (n) should be stated in the figure title or legend.
Line 243-244: Are the 931 inside t(931) the sample size? But line 168 said 933 participants. It is surprising that a score difference of 0.44 (i.e. 15.41 - 15.85 = -0.44) with huge error bars could lead to a p value of 0.012. Are you certain that the correct statistics test have been employed. Also, the females score slightly lower here, but your next sentence (line 245-246) said the female scored higher.
Line 245-246: Some new paragraphs are not indented.
Line 251-252: Similar to the problems in Line 243-244, the scores of the two groups were virtually the same (23.50 vs. 23.03, with a huge margin of errors) but the p-values were very significant (P-value of 0.017). Perhaps, to show the validity of the statistical analysis, would you please provide the editor and reviewers the raw data and the statistical method to demonstrate how the statistical significance could be generated – even though this does not need to be published. For this specific comparison, which statistical test was applied. Same for the rest of the statistical comparison in this paragraph.
Discussion
Line 271-277: This paragraph seems to be a re-introduction of functional constipation rather than discussion of your study data. Consider moving it to the introduction paragraph.
Line 280-324: The discussion paragraphs tried to associate the findings with parents’ knowledge. But the current study was not exclusively on parents.
Conclusion
Line 333: This study was not exclusively on parents. Therefore, the authors may not be able to conclude the study showed local parents’ perception.
Line 337-346: The study showed the study participants’ knowledge but did not demonstrate the benefit of national health program, social media, public educatory tools, and school programs. It may be a far stretch to make such a big conclusion based on the current study data.
References
Line 378-446: Not sure whether the month of the journal article publication is needed for this Int. J. Environ. Res. Public Health journal. If not, please remove all of them.
Comments on the Quality of English LanguageThere are some grammatical and syntax errors detected. Some new paragraphs are not indented. Please see specific comments. I can only point out some of them.
Author Response
Thanks for sending the detailed review comments for this manuscript to us to revise for consideration again. We wish to thank you for your comments and suggestions. We have responded to the listed comments in the attached file below. We submitted the updated file under the Upload Revised Manuscript section.

Round 2
Reviewer 1 Report
Comments and Suggestions for Authors
The authors have addressed all of my concerns satisfactorily, and the manuscript can now be accepted for publication.
Author Response
We value your guidance and support and are glad about your decisions. I have attached the approval of our manuscript from a proofreading expert.

Reviewer 2 Report
Comments and Suggestions for Authors
The authors have revised the manuscript accordingly. It can be considered for publication.
Author Response

(The authors gave the same response as above.)

Reviewer 4 Report
Comments and Suggestions for Authors
N/A
Comments on the Quality of English LanguageN/A
Author Response
Dear respected Reviewer,
Thanks for sending the detailed review comments for this manuscript to us to revise for consideration again. We wish to thank the reviewers for their comments and suggestions. We have responded to the listed comments as below.
General comments:
1- It is unclear what the clinical significance of this study is. It needs to be clear what new knowledge this study is bringing to the scientific community. E.g. what are the big picture the authors want to tell us now that they found that there are difference in knowledge scores in different age and gender. The authors also commented they found no distinct difference between the current study and similar national studies (line 327-328). Then, what is innovative about this study?
Answer: Thanks for the comment. According to your comments, we modified and highlighted the changes in yellow with red font (Line 357-362). Please you can see the modification in the following:
The people with multiple children between 20 and 29 years old who had completed their education up to and including a bachelor's degree gave the most accurate answers regarding their knowledge, practice, and beliefs. This further emphasises the significance of our research, as it demonstrates that the correct knowledge, practice, and belief are essential for managing constipation, a discovery that has not been documented in previous studies.
2- The authors stated in the introduction (line 70-73) that the study aimed to empathize on understanding the challenges parents face in managing the condition and its impact on the children’s quality of life. But the study data seemed to be only focusing on the participants’ level of knowledge. The abstract also stated the study data represented the views of the parents of the local community. But there is a lack of description of what local community this study was investigating.
Answer: We modified it according to your comment. We added the following sentences in the abstract, the survey received 933 responses. The target population was adults over 18 years of age living in the Eastern Province of Saudi Arabia.
We modified the following sentences in the introduction, The prevalence of FC varies widely between countries and studied regions, and few studies assess the parents’ perception of FC in the Eastern Province of Saudi Arabia. Some studies in other countries related the problem to parents’ psychological issues [21]; in local studies, they constrained more on their knowledge of diet causes [22]. Examining the literature will reveal that certain causative factors are contingent upon the local community’s beliefs. Therefore, this study aimed to explore the local community’s view of FC, focusing on understanding the challenges faced in managing the condition and its impact on the children’s life quality in the Eastern Province of Saudi Arabia.
3- Some of the participants here were apparently not parents as they declared having no children (n = 150) in Table 4 (line 174). But author’s study aim (line 70-73) stated the study aim was to explore the local parental view of functional pediatric habitual constipation. This could be a serious flaw in this study as the results need to be supportive of the study aim and conclusion. If the authors want to maintain their objective of exploring the local parental view, major revisions are needed. E.g. removing the 150 participants from the 933 total participants and re-analyzed all the data.
Answer: Thanks for the comment. The aim was to explore the knowledge, perception, and practice toward FC among all adults planning to have a child and living in the study area. However, we appreciate that you added the following sentences in the limitation(line 368-370): the study has a relatively limited number of participants compared to the whole population. Additionally, some parents lack experience with children, which may introduce biases in the research results.
4- Some of the statistical comparisons were very intriguing in which the two comparison groups had virtually the same scores, but generated a significant p-value. E.g. Line 251-252 (15.41 vs. 15.85 with huge error bars but generated a p value of 0.0120; Line 243-244 (23.50 vs. 23.03, with huge error bars, but the p-values were very significant at 0.017. Although the raw data does not need to be published, perhaps, this should be submitted to the reviewers and editor, with description of the statistical test used, to demonstrate the validity of these statistical comparison.
Answer: Thanks for the suggestion. According to your comments, we modified it with red font (Line 261-286). Please you can see the modification in the following:
The minimum score for the ‘Knowledge about constipation’ was 9, the maximum was 20, and the mean score was 15.76 ± 2.02. The independent samples t-test indicat-ed a statistically significant difference in mean knowledge scores between males (15.41 ± 2.25) and females (15.85 ± 1.94) [t (931) = −2.701, P = 0.007] with a small ef-fect size ( Cohen’s D = −0.21). Specifically, females achieved a significantly higher mean knowledge score compared to males. Furthermore, ANOVA was conducted to examine the relationship between the number of children groups and knowledge scores. The results revealed a statistically significant difference in mean scores [F (3,929) = 5.773, P = 0.001] with a small effect size ( η² =0.011). The results of the Bon-ferroni post hoc test indicated that participants who had three or more children (15.98 ± 2.18) exhibited significantly higher knowledge scores than those with only one child (15.23 ± 1.99) (P = 0.001).
The ‘Knowledge about feeding’ had a minimum score of 17, a maximum of 30, and a mean of 23.21 ± 2.05. A statistically significant difference was observed among different age groups, as determined by ANOVA [F (2,930) = 4.023, P = .018], with a small effect size (η² = 0.009). The results of the Bonferroni post hoc test indicated that individuals within the age range of 20–29 years (23.50 ± 2.09) exhibited significantly higher knowledge scores than those between 30–39 years (23.03 ± 2.3) (P = 0.017).
The ‘Perception about FC‘ scores displayed a minimum of 4, a maximum of 20, and a mean score of 14.76 ± 2.84. ANOVA also revealed a statistically significant dif-ference in the mean scores of ‘Practice of toilet training’ among groups with different educational levels [F (3,929) = 5.295, P = 0.001], with a small effect size (η² = 0.017). The Bonferroni post hoc test indicated a significant difference in practice scores be-tween participants who possessed a postgraduate degree (15.45 ± 2.75) and those with less than a Bachelor’s degree (14.22 ± 3.25), the former group exhibiting higher scores (P = 0.001). The ‘Practice of toilet training’ score ranged between 12 and 30, with a mean score of 23.03 ± 2.86. No significant differences were observed in the mean scores among the different respondent groups.
Specific comments:
1- Abstract
Line 15-18: Generally speaking, the abstract should describe at least the sample size and setting (Saudi Arabia - which local community?).
Answer: We modified that according to your suggestion. We added the following sentences in the abstract (line 17-19), the survey received 933 responses. The target population was adults over 18 years of age living in the Eastern Province of Saudi Arabia.
2- Abstract
Line 18: Perhaps, it is best to clarify whether the females or males scored higher to prevent confusion and misinterpretation.
Answer: Thanks for the suggestion. We modified that according to your suggestion. We added the following sentences in the abstract (line 20-21), the mean knowledge scores were significantly higher in females than males
3- Abstract
Line 23-26: It said the local community already have a good perceived knowledge about functional pediatric habitual constipation. Then why is it important to explore the local community view of constipation among children?
Answer: Thank you for your comment. We added some clarification (in line 25-26).
This study demonstrated that parents of the local community have a good perceived knowledge about FC, but it needs to be linked with the practice.
4- Introduction
Line 40-41: Consider using colon rather than semicolon after “The criteria is” e.g. “The criteria are the followings: “
Answer: We modified it according to the reviewer's comment.
5-Introduction
Line 55-56: The part about “remain suffering” need to be more concise. E.g. how long are they remaining to suffer? What are they suffering on? Psychological or physical suffers?
Answer: We modified it according to your comment in line 54-60.
Paediatric constipation constitutes a scope of different organic and functional eti-ologies [12]. Its origin varies according to the child’s age from the neonatal period to adolescence [13]. The passage of meconium is a crucial marker for newborn health; any delay should be promptly investigated. Sometimes, it may indicate slow transit con-stipation [14] and is an indicator of other abnormal colonic diseases, including malro-tation, colon atresia, or Hirschsprung’s disease, which can be challenging to manage and require surgical intervention [15].
6- Introduction
Line 61: “This” often requires pharmacological intervention. Can you clarify what “this” means?
Answer: We modified it according to the reviewer's comment in line 66-68.
Those children’s symptoms can be effectively managed by combining non-pharmacological and pharmacological interventions (i.e., Osmotic Laxatives, polyethylene glycol, lactulose, stimulant laxatives, bisacodyl, and sodium picosulfate)
7- Introduction
Line 62-64: Because guidelines from different areas differ. When you talk about first- and second-line treatment, it is best to be explicit on which guidelines recommend that, rather than having a reference only. It seems like reference 16 is based on a literature review conducted by two authors only, but not from an authoritative guideline.
Answer: Thank you for your comments. We try to explore a brief about the treatment of Constipation from literature.
8- Introduction
Line 69-70: “When we look through these literatures, we will find that some of the causal factors are variable according to local community beliefs.” I am not sure what purpose this sentence serves. Or are you saying you have already looked at these factors and now want to summarize the findings? If this sentence does not add anything, please consider removing it.
Answer: Thanks for the suggestion. We modified that according to the reviewer suggestion.
9- Methods
Line 76: consider using all lower case for “Practice”
Answer: Thanks for the comment. We modified that according to your comment.
10- Methods
Line 80: what is the “at the end of the sentence for? It is important to reference or mention which sample size calculation you used. It is unclear why you said a population size of 5 million but a minimum sample size of 385.
Answer: Thank you for your comments. We added the following clarification in lines ( 85-95).
The sample size for a finite population (5000,000 residents) was determined using the World Health Organization (WHO) sample size calculator. An acceptable error rate of 5% and an expected proportion of awareness in the population of 0.5 were considered. Additionally, a Type I error rate of 5% (α=0.05) was used. The calculated was sample size was 385, However 933 participants responded to the survey.
- WHO. sample-size-calculator. https://cdn.who.int/media/docs/default-source/ncds/ncd-surveillance/steps/sample-size-calculator.xls?sfvrsn=ee1f4ae8_2.
11- Methods
Line 81: To enhance the reproducibility of a study, the methodology can include the original Arabic language questionnaires and their English translation in the Appendix.
Answer: We added your request to the supplementary section.
12- Methods
Line 83: How are the target population (parents) reached? Do you mean all the school and preschool parents in Saudi Arabia, or only in a certain region? At a minimum, the setting of the research needs to be described. Were there rewards / compensation given to the participants? What are the inclusion and exclusion criteria for participants?
Answer: The target population was adults above 18 years living in the eastern province of Saudi Arabia. The participation was voluntary.
13- Methods
Line 90: The part about literature review to identify items seem a bit vague. Do you mean you adapted the survey questions from references 18-23?
Answer: Yes, after reviewing the cited references we developed our questionnaire items from them.
14- Methods
Line 95-98: The LAWSHE method and Content Validity Ratio should be referenced in case readers want to check the validity of the methods.
Answer: Thank you for your comment. We added the Reference in line 106.
Lawshe CH. A quantitative approach to content validity. Personnel psychology. 1975 Dec 1;28(4):563-75.
15- Methods
Line 103-111: Similarly, the Exploratory factor analysis, Kaiser–Meyer–Olkin test, oblique rotation method (Oblimin), and Principal Axis Extraction method should be referenced.
Answer: Thank you for your comment. We added the Reference in line 116.
Fabrigar LR, Wegener DT. Exploratory factor analysis. Oxford University Press; 2011 Dec 22.
16- Methods
Line 138: The Fornell and Larcker criteria should be referenced.
Answer: Thank you for your comment. We added the Reference in line 148.
Rasoolimanesh SM. Discriminant validity assessment in PLS-SEM: A comprehensive composite-based approach. Data Analysis Perspectives Journal. 2022 Feb;3(2):1-8.
17- Results
Line 171: Perhaps, for the readers who are unfamiliar with the Saudi Arabia education system, can you explain what “uneducated” mean (either in results or methodology)? Do you mean no sort of education whatsoever since childhood?
Answer: Thank you for your comment. We mean they are not educated.
18- Results
Line 174: “one, two, and more” should have the first letter capitalized like the other rows
Answer: Thanks for the suggestion. We modified that according to your suggestion.
19- Results
Line 174: Some of the participants here were apparently not parents as they declared no children. But author’s study aim (line 70-73) stated the study aim was to explore the local parental view of functional pediatric habitual constipation.
Answer: We modified that according to your suggestion. We answer this question in general question number 3.
20- Results
Line 185: Why is there a semicolon after “whether”?
Answer: We have Removed
21- Results
Line 190 (Table 5): It is difficult for readers to visualize what is supposed to be the correct response in each question.
Answer: True answers are in red colour and italics, mentioned under the table.
22- Results
Line 199-200 (Figure 3): Generally speaking, a figure in a journal should be self-explanatory on its own. Are these numbers percentages? The number of respondents (n) should be stated in the figure title or legend.
Answer: It is corrected to “The percentage of participants ’anticipations for the different causes of constipation (N=933).”
23- Results
Line 214 (Table 6): Like the problem in Table 5, it is difficult for readers to visualize what is supposed to be the correct response in each question. Please consider remaking this entire label for clarity. It also means some rewriting of Lines 201-212 for clarity.
Answer: True answers are in red colour and italics, mentioned under the table.
24-Results
Line 218: Why is there a comma after “allowed”?
Answer: We have Removed.
25- Results
Line 232 (Table 7): Like the problems in Table 5 and 6, what is supposed to be the correct response here? How are these responses relevant to functional habitual constipation?
Answer: Good practices are in red and italics, mentioned under the table.
26- Results
Line 239-240 (Figure 4): Like line 199-200 (figure 3): Are these numbers percentages? The number of respondents (n) should be stated in the figure title or legend.
Answer: Thank you for your comment. We have corrected it.
27- Results
Line 243-244: Are the 931 inside t(931) the sample size? But line 168 said 933 participants. It is surprising that a score difference of 0.44 (i.e. 15.41 - 15.85 = -0.44) with huge error bars could lead to a p value of 0.012. Are you certain that the correct statistics test have been employed. Also, the females score slightly lower here, but your next sentence (line 245-246) said the female scored higher.
Answer: - t(931) is degrees of freedom for the t test.
- the actual p value is 0.007 , the written one (0.012, which is the p value for levene test of equality of variance) is by mistake.
- females scored higher than males à corrected.
- Although the mean difference is -0.44 but it was statically significant with small effect size (0.21) and this could attributed to the large sample sizeà ( Sullivan GM, Feinn R. Using Effect Size-or Why the P Value Is Not Enough. J Grad Med Educ. 2012 Sep;4(3):279-82. doi: 10.4300/JGME-D-12-00156.1. PMID: 23997866; PMCID: PMC3444174.).
- corrections were made.
28- Results
Line 245-246: Some new paragraphs are not indented.
Answer: We have corrected.
29- Results
Line 251-252: Similar to the problems in Line 243-244, the scores of the two groups were virtually the same (23.50 vs. 23.03, with a huge margin of errors) but the p-values were very significant (P-value of 0.017). Perhaps, to show the validity of the statistical analysis, would you please provide the editor and reviewers the raw data and the statistical method to demonstrate how the statistical significance could be generated – even though this does not need to be published. For this specific comparison, which statistical test was applied. Same for the rest of the statistical comparison in this paragraph.
Answer: - for large sample size its expected to have such significant diffrence although the effect size is small. à( Sullivan GM, Feinn R. Using Effect Size-or Why the P Value Is Not Enough. J Grad Med Educ. 2012 Sep;4(3):279-82. doi: 10.4300/JGME-D-12-00156.1. PMID: 23997866; PMCID: PMC3444174.).
30- Discussion
Line 271-277: This paragraph seems to be a re-introduction of functional constipation rather than discussion of your study data. Consider moving it to the introduction paragraph.
Answer: Thanks for the suggestion. We modified that according to the reviewer suggestion.
31- Discussion
Line 280-324: The discussion paragraphs tried to associate the findings with parents’ knowledge. But the current study was not exclusively on parents.
Answer: Thanks for the suggestion. We modified that according to your comment suggestion ( line 297-323).
32- Conclusion
Line 333: This study was not exclusively on parents. Therefore, the authors may not be able to conclude the study showed local parents’ perception
Answer: Thanks for the suggestion. We modified that according to the reviewer's suggestion.
33- Conclusion
Line 337-346: The study showed the study participants’ knowledge but did not demonstrate the benefit of national health program, social media, public educatory tools, and school programs. It may be a far stretch to make such a big conclusion based on the current study data.
Answer: We are trying to have more education regarding that.
34- References
Line 378-446: Not sure whether the month of the journal article publication is needed for this Int. J. Environ. Res. Public Health journal. If not, please remove all of them.
Answer: Thanks for the suggestion. We modified that according to the reviewer's suggestion.
